# Full-Thickness Compressive Corneal Sutures with Removal of Anterior Chamber Air Bubble in the Management of Acute Corneal Hydrops

Zahra Ashena [1], Ritika Mukhija [1] and Mayank A. Nanavaty [1,2,*]

1   Sussex Eye Hospital, Brighton & Sussex University Hospitals NHS Trust, Eastern Road, Brighton BN2 5BF, UK
2   Brighton & Sussex Medical School, University of Sussex, Falmer, Brighton BN1 9PX, UK
*   Correspondence: mayank.nanavaty@nhs.net

**Abstract:** Acute hydrops is a rare complication of corneal ectatic disease, which occurs secondary to Descemet membrane break. Spontaneous resolution of this condition is associated with longstanding ocular discomfort and corneal scar. Intracameral gas/air injection with or without corneal suturing, anterior segment ocular coherence tomography (ASOCT)-guided drainage of intrastromal fluid, and penetrating keratoplasty are some of the described surgical interventions to manage this condition. The purpose of our study was to assess the effect of full-thickness corneal suturing as a solo treatment in the management of acute hydrops. A total of five patients with acute hydrops received full-thickness corneal sutures perpendicular to their Descemet break. A complete resolution of symptoms and corneal oedema was observed between 8 to 14 days post-operation with no complications. This technique is simple, safe, and effective in the management of acute hydrops and saves patients from a corneal transplant in an inflamed eye.

**Keywords:** full thickness suturing; keratoconus; hydrops

## 1. Introduction

Acute corneal hydrops is a visually debilitating complication of ectatic corneal diseases, which results from a tear in the Descemet's membrane (DM) followed by the entry of aqueous humour into the corneal stroma [1]. Vernal keratoconjunctivitis (VKC), steeper keratometry, atopy, Down syndrome, and eye rubbing are important risk factors for corneal ectasia such as keratoconus where corneal hydrops can develop [2,3].

Patients present with intense photophobia, pain, and reduced visual acuity due to significant corneal oedema [4]. Formerly, the management of hydrops was often conservative as it can resolve spontaneously with extensive scarring. Medical treatment, including topical steroids, antibiotics, cycloplegics, hypertonic saline, ocular antihypertensive, and lubricants, aims to reduce inflammation and provide symptomatic relief. However, spontaneous resolution of corneal hydrops takes longer (up to 36 weeks) [5], and persistent oedema causes prolonged discomfort and complications such as infection, scarring, neovascularization, and permanent visual loss [6,7].

Some interventions have been described in the literature to accelerate the resolution of hydrops and hasten visual recovery. These include injection of intra-cameral air or gas with or without compressive sutures, pre-Descemet suturing [8], anterior segment ocular coherence tomography (ASOCT)-guided drainage of intra-stromal fluid [9,10], amniotic membrane transplant [11], mini-Descemet's membrane endothelial keratoplasty (DMEK) [12], penetrating keratoplasty, and, in the rare event of corneal perforation, application of tissue adhesives [6]. We aim to describe the outcomes of full-thickness compressive corneal suturing as a primary and sole treatment in managing acute corneal hydrops.

## 2. Methods/Surgical Procedure

All patients had ASOCT at presentation to locate the Descemet tear. However, this diagnostic tool was not conclusive in a few cases with significant corneal oedema and stromal cleft. After obtaining informed consent, the theatre procedure was performed under local or general anaesthesia, based on the patient's preference. A small air bubble was injected into the anterior chamber to visualize the Descemet tear, which revealed the break/s and demonstrated their orientation and length within 10 min. This technique was especially beneficial in cases with extensive oedema, where ASOCT failed to locate the Descemet break. This was followed by placing 3 to 6 full-thickness interrupted 10-0 nylon corneal sutures perpendicular to the Descemet break/s (Video S1, Figure 1a–f). Care was taken not to touch the crystalline lens. The sutures were tied tighter than usual, expecting some loosening on the resolution of corneal oedema and the knots were buried in the stroma. The small air bubble was removed at the end of the procedure, and intracameral cefuroxime was injected. The eyes were covered with a clear shield. Patients were prescribed topical Tobramycin and Dexamethasone combination eye drops (Tobradex®, Alcon Laboratories, Fort Worth, Texas, USA) four times a day for two weeks, and a follow-up was arranged in one week.

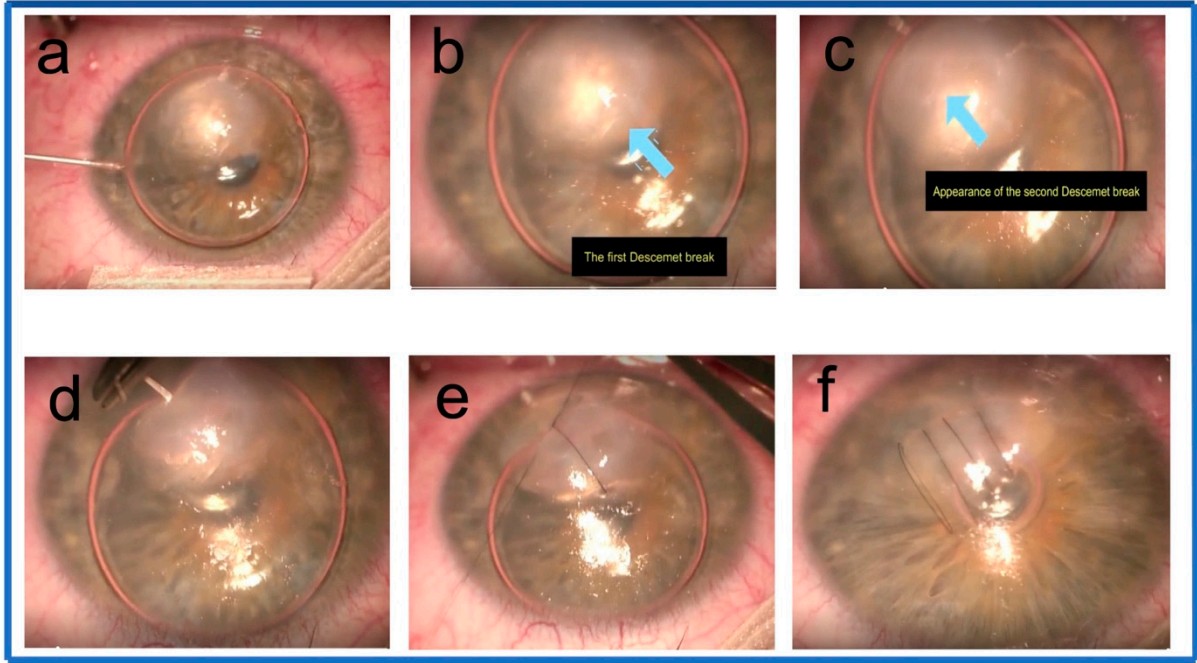

a. Injecting air bubble in the AC via paracentesis, b. and c. appearance of two Descemet breaks, d. placing the perpendicular full-thickness suture, e. making the suture tight, f. four full-thickness suture perpendicular to the Descemet breaks

**Figure 1.** (**a**) Injecting the air bubble in the AC via paracentesis, (**b**,**c**) the appearance of two Descemet breaks, (**d**) placing the perpendicular full-thickness suture, (**e**) making the suture tight, and (**f**) four full-thickness suture perpendicular to the Descemet breaks.

## 3. Report of the Cases

Case 1: A 23-year-old Caucasian male, known keratoconus since the age of 17, presented with acute hydrops in his left eye. His uncorrected visual acuity (UCVA) was counting fingers (CF) on examination. He received four full-thickness corneal compressive sutures under general anaesthesia (GA). The procedure helped provide significant symptomatic relief and reduced his apex pachymetry on OCT from 1096 μ to 466 μ in 10 days (Figure 2a). A total of four weeks later, the sutures were loose and hence removed (Figure 2b). Two months after the initial presentation, his UCVA was CF with further

improvement to 6/18 with pinhole, similar to his pre-hydrops visual acuity. He was not keen on rigid gas permeable contact lenses (RGPCL) since he had a good unaided vision in his fellow eye.

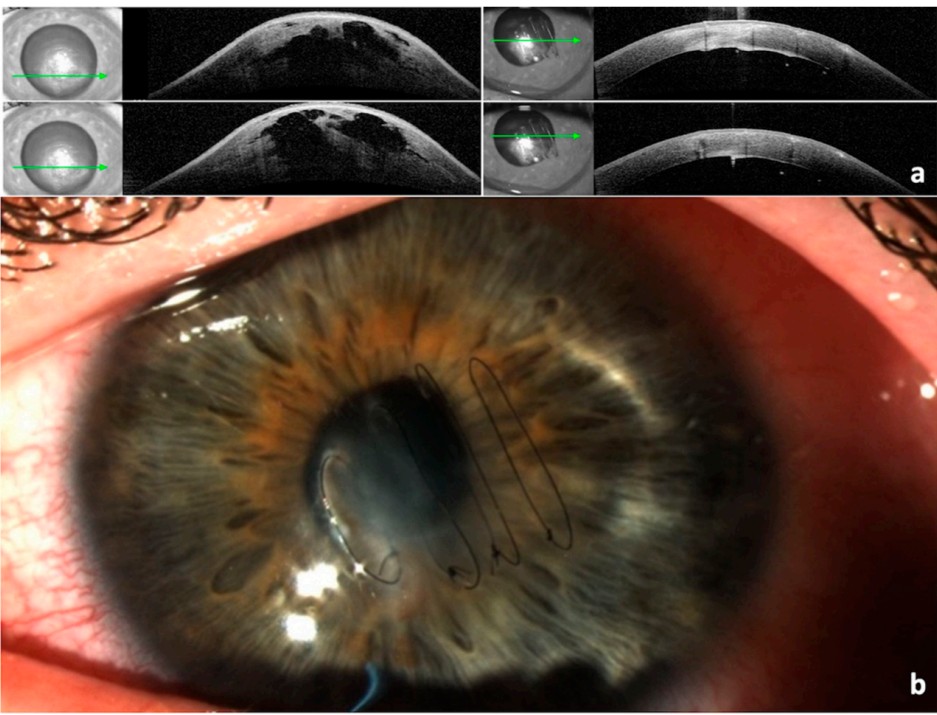

**Figure 2.** (**a**) Anterior segment OCT: pre-op (**left**) with apex pachymetry of 1096 μ and 10 days post-op (**right**) with apex pachymetry of 466 μ. The arrow shows the level of the scan. (**b**) Loose corneal sutures after resolution of corneal oedema.

Case 2: A 20-year-old female of Afro-Caribbean origin, diagnosed with keratoconus since the age of 18, presented with acute hydrops in her right eye. Her best spectacle-corrected visual acuity (BSCVA) was hand movement (HM). Her maximum keratometry increased from 83 dioptres (D) to 124D, and her apex pachymetry changed from 340 μ to 1380 μ. ASOCT confirmed the presence of stromal clefts (Figure 3). Conservative management was started, and within one week she received four full-thickness compressive corneal sutures, as described above. There was a complete resolution of her symptoms and corneal oedema in 8 days. The sutures were removed in four weeks. She was not keen on RGPCL. Her final BSCVA improved to 6/36 in the affected eye, one line lower than her pre-hydrops vision of 6/24.

Case 3: A 41-year-old male of South Asian origin, with a strong history of atopy and known keratoconus since the age of 19, presented with acute hydrops in his left eye. His best-corrected visual acuity (BCVA) from 6/9 with scleral contact lenses pre-hydrops dropped to hand movement. His apex pachymetry increased to 1473 μ from 398 μ. ASOCT showed large stromal clefts (Figure 4a). He received six full-thickness compressive corneal sutures perpendicular to his Descemet break two days after presentation. The compressive sutures significantly helped with his symptoms and corneal oedema (Figure 4b) and his apex pachymetry decreased to 532 μ in two weeks. His corneal sutures were removed in 4 weeks, and his final BCVA, using a scleral contact lens, was CF three months from hydrops.

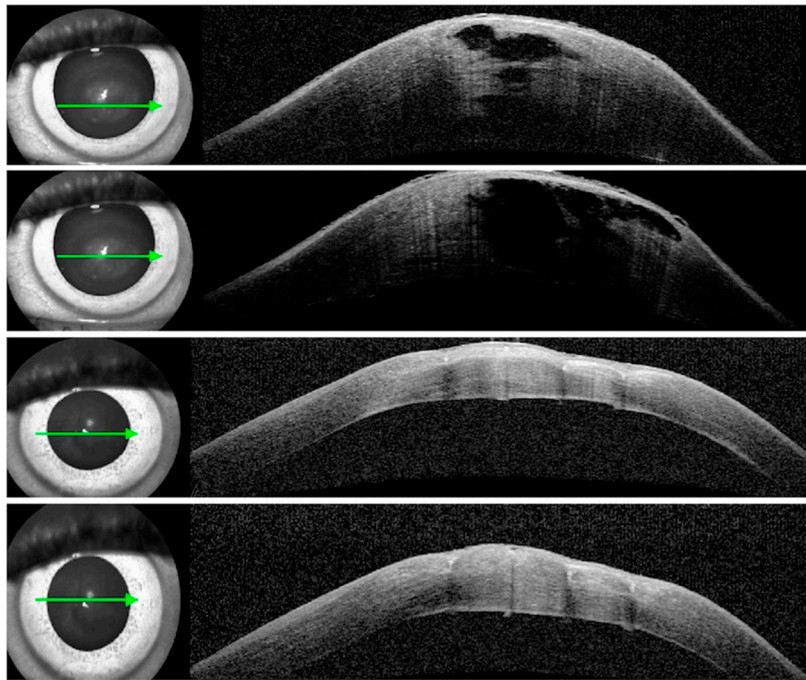

**Figure 3.** Anterior segment OCT: pre-op with a stromal cleft (**top** 2 scans) and apex pachymetry of 1380 μ, and 8 days post-op (**bottom** 2 scans) with apex pachymetry of 548 μ. The arrow shows the level of the scan.

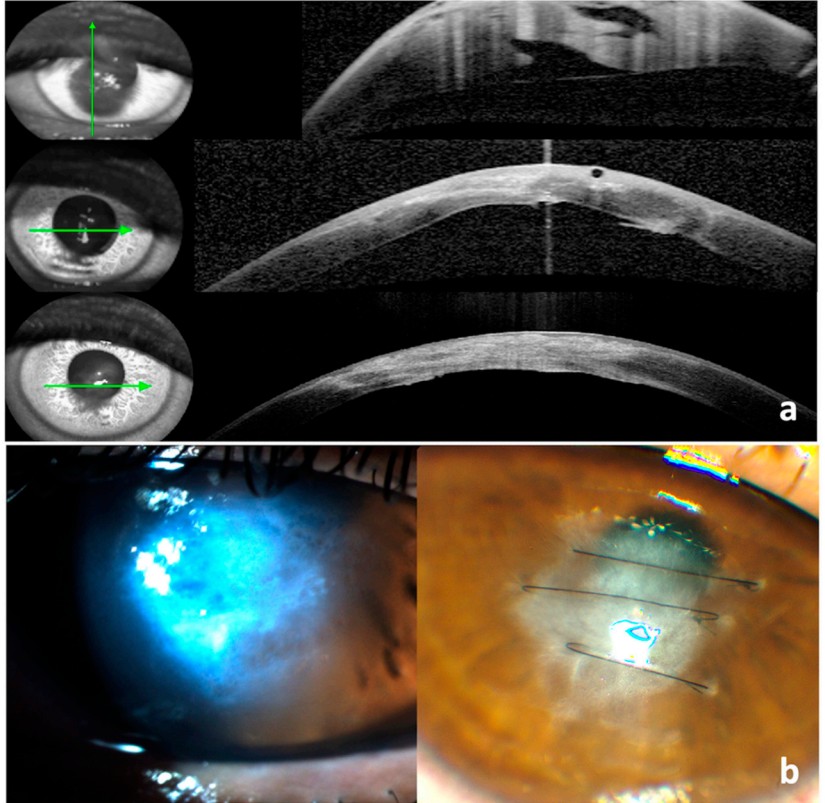

**Figure 4.** (**a**) Anterior segment OCT: pre-op (**top**) with apex pachymetry of 1473 μ, 15 days post-op (**middle**) with apex pachymetry of 532 μ, and 8 weeks post-op (**bottom**) with apex pachymetry of 402 μ; (**b**) Anterior segment photo: pre-op (**left**) and 15 days post-op (**right**). The arrow shows the level of the scan.

Case 4: A 36-year-old Caucasian male, diagnosed with keratoconus at 27 years of age, presented with right eye acute hydrops, where conservative treatment was initiated. He was thereafter reviewed for a follow-up in one month in the cornea clinic when he was still symptomatic with pain and photophobia. On assessment, ASOCT showed significant stromal oedema and his BCVA had dropped from 6/9 with scleral contact lenses pre-hydrops to counting fingers (CF). His apex pachymetry measured 1194 μ. He was offered compressive corneal suturing perpendicular to his Descemet break. In nine days, his apex pachymetry decreased to 430 μ. The corneal sutures were removed five weeks later (Figure 5a), and his BCVA with scleral contact lenses improved to 6/9, which was similar to his pre-hydrops vision.

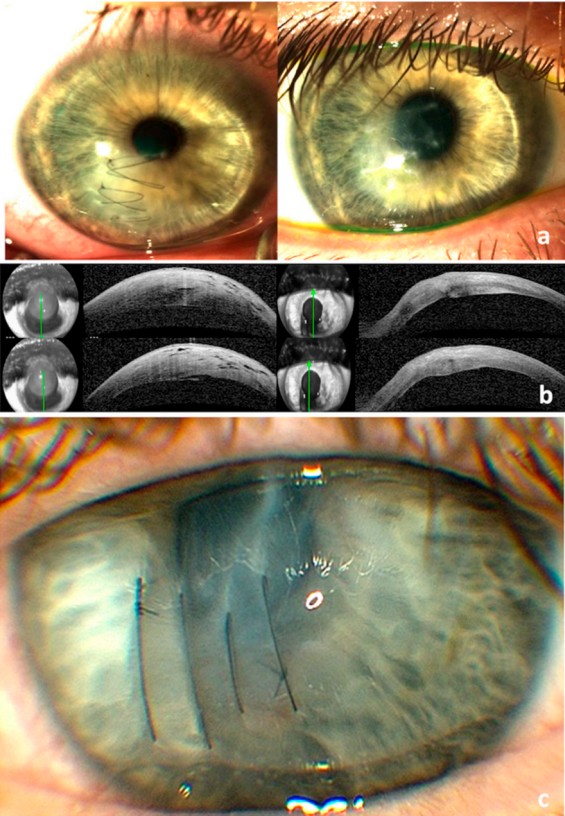

**Figure 5.** (**a**) Anterior segment photo of the right eye 4 weeks post-suturing (**left**) and after suture removal (**right**); (**b**) Anterior segment OCT: pre-op (**left**) with apex pachymetry of 1024 μ and 14 days post-op (**right**) with apex pachymetry of 493 μ; (**c**). Anterior segment photo of left eye post-suturing.

Case 5: A 41-year-old Caucasian male, known with keratoconus since the age of 22 years, presented with left corneal hydrops. His VA at presentation was HM with no improvement with pinhole, and his apex pachymetry was 1024 μ. He received four compressive sutures to his Descemet break seven days from the presentation. Two weeks post-procedure, his discomfort had settled, and apex pachymetry was measured at 493 μ (Figure 5b,c). Since then, he has returned to his local hospital, and we have no further information on his visual outcome.

## 4. Discussion

Conventional management of acute hydrops mainly focused on providing symptomatic relief. This included topical sodium chloride eye drops to decrease corneal oedema, cycloplegic eye drops to address the photosensitivity, bandage contact lenses to help with the pain, aqueous suppressants to lower the intraocular pressure and encourage corneal dehydration, topical steroids and non-steroids to alleviate the inflammation, and topical antibiotics to prevent the secondary infection [1,4,6]. However, recovery from acute corneal

hydrops may take a few months without surgical intervention. This is associated with prolonged pain, photosensitivity, stromal vascularization, and inflammation and can lead to corneal scaring and worse visual outcomes. Emergency penetrating keratoplasty used to be an alternative surgical option to manage acute hydrops [1]. However, this is not the recommended treatment at present as no report has been published on the survival of penetrating keratoplasty in acute hydrops, and corneal transplant in an inflamed eye is associated with a significantly higher risk of rejection and failure [13].

A few surgical interventions have been described to hasten the recovery in acute corneal hydrops. Miyata et al. tried intracameral air injection and although the final visual outcome was similar between their intervention and control group, a significantly quicker resolution of corneal oedema was observed in the intervention group, but a few injections were required [14]. Shah et al. used perfluoropropane (C3F8) to save patients from repeated intracameral air bubble injections. Two intracameral injections of non-expansile C3F8 gas were given five days apart, and complete resolution of corneal oedema was noticed shortly after the second injection without any complications [15]. Basu et al. conducted a comparative study on using non-expansile C3F8 gas to manage acute hydrops in 62 eyes and compared the outcome with 90 eyes in the control group. The resolution of symptoms and corneal oedema was significantly quicker in the intervention group, with reversible pupillary block and elevated IOP being the main complication of their intervention. No significant difference was noticed between the groups regarding the final visual outcome [16]. Rajaraman et al. reported outcomes of combined C3F8 and compressive sutures in managing acute hydrops in 17 patients with keratoconus. They noticed that intracameral C3F8 alone was not enough to resolve corneal oedema in the presence of stromal clefts. To prevent pupillary block and glaucoma, they applied immediate post-op pupillary dilation, and no complication was reported in their series of patients[2]. Cherif et al. introduced the pre-Descemet sutures to manage Descemet detachment due to corneal hydrops [8]. This technique was combined with intra-cameral air injection and proved safe and effective in shortening the hydrops and resolving oedema. The challenge with this technique is limited visualization of the cornea and pre-Descemet area in the presence of significant stromal oedema. To overcome this challenge, further modifications were suggested, such as full-thickness compressive suture instead of pre-Descemet suture or the use of intra-operative OCT (iOCT) to visualize the pre-Descemet cornea. Mohebbi et al. tried the combination of full-thickness compressive suture and injection of SF6. They found this method very effective and safe, with complete resolution of corneal oedema in 5–24 days without any complication [3]. OCT-guided drainage of intrastromal fluid, using venting incisions combined with intracameral air injection, has also been tried [9]. This is reported to achieve Descemet attachment on the first day post-op with complete resolution of corneal oedema within 2–3 weeks. Later, iOCT was utilized in another study to remove intra-stromal fluid pockets and place pre-Descemet sutures in two patients with acute hydrops. The authors injected intracameral SF6 to tamponade the detached Descemet and achieved good results [10]. Venting incisions to compress the stromal fluid combined with intracameral injection of 20% SF6 was successfully tried in another study [17].

Bachmann et al. reported a mini-DMEK technique to treat three patients with acute hydrops [12]. The procedure was successful in two patients and needed repeating in one patient. To apply this technique, iOCT is necessary due to the poor visibility, a donor graft is required on an urgent basis, and the AC needs to be filled with gas. While iOCT is not readily available in all centers, and donor tissue shortage is another concern in many countries, and the use of anterior chamber gas can be associated with potential complications such as pupillary block and cataract.

Our study reports the full-thickness compressive corneal suturing as a solo treatment in managing acute hydrops. This is a relatively simple technique that surgeons can easily adopt. Furthermore, although iOCT is an excellent tool for these cases, this technique could be performed in absence of iOCT since the sutures are full-thickness. The key is to

visualize the Descemet breaks using a small anterior chamber air bubble before suturing. This enables the surgeon to place the sutures perpendicular to the break.

Most of the previous reports combined this technique with the injection of air or gas into the anterior chamber. Although some studies with a small number of patients did not come across the pupillary block [3], others with larger sample size noticed up to 16% rate of pupillary block in their patients [16], as it is already understood very well [18]. To prevent this complication, either prophylactic peripheral iridotomy (PI) is to be performed, which is not only difficult to perform via oedematous cornea with poor visibility but also associated with potential complications such as hyphaema [19], IOP spike [19], dysphotopsia [20], and similar, or the need for topical cycloplegic drops post-operatively, as long as the gas is present in the AC. However, with our technique, there is no requirement for prophylactic PI or post-op cycloplegic agents. In all five reported cases, we achieved good anatomical outcomes without leaving any air bubbles in the anterior chamber.

Moreover, although opacification of the crystalline lens was not reported in the previous reports where they used air/gas bubbles to manage acute hydrops, we do not have detailed information on the duration of gas remaining in the anterior chamber and the duration of post-procedure follow-up in those reports. The posterior segment gas bubble induces cataracts [21], and several reports have been published on IOL opacification with anterior chamber air/gas bubble following endothelial keratoplasty [22]. Considering the young age of these patients and further challenges in intraocular lens power calculation and visual rehabilitation, it is best to avoid leaving any air in the anterior chamber of these patients, which can potentially induce cataract.

Regarding the visual outcome, post-procedure BCVA, compared to the BCVA at presentation, improved in four patients and was unknown in one patient who did not attend the follow-up. Two of our patients achieved their pre-hydrops visual acuity; one patient experienced a one-line drop on the Snellen chart, and another patient, with extensive involvement of the central cornea, had a significant drop on the Snellen chart. Cherif et al. also compared the BCVA at presentation with post-suturing BCVA, which showed a significant improvement [8]. Furthermore, two of their patients required PK once the hydrops settled. Although the visual outcome is not provided in all patients, the graft patients achieved BCVA of 20/20 and 20/32 and another non-grafted eye achieved BCVA of 20/20 [8]. Therefore, the 6-month post-suturing BCVA in their study reflects not only the suturing outcome but also the added PK benefit [8]. Likewise, the visual outcome in the mini-DMEK technique is not comparable to our study as they provided unaided visual acuity in their three patients, which is understandably poor in eyes with keratoconus regardless of hydrops [12]. Nevertheless, a critical factor determining the final visual outcome is the extent and location of DM break and corneal oedema. The involvement of the central cornea will have a remarkable effect on the final visual outcome. In contrast, paracentral or peripheral DM break and oedema's impact on final vision will be less significant. Some previous studies reported no significant difference in visual outcomes between their intervention and control groups [14,16]. Instead of comparing the final visual acuity between two groups, we suggest comparing each patient's pre-hydrops and post-treatment/observation visual acuity to provide a more accurate result.

Although we did not encounter any complications when placing full-thickness corneal suture, care should be taken not to damage the adjacent structures. Also, to avoid infection, the use of intracameral antibiotics is highly recommended. Placing corneal sutures can also be associated with suture-tract linear scarring. This was noticed with both pre-Descemet [8] and full-thickness suturing [3]. We also came across this complication, however, similar to the previous studies, it did not appear to cause a significant visual problem, and good vision was achieved with RGPCL.

In summary, full-thickness corneal suturing as a solo technique in managing acute hydrops is effective and safe. This procedure can be performed without iOCT and does not necessitate techniques to tackle pupillary block or raised IOP. This technique prevents prolonged discomfort and corneal oedema, leading to vascularization and scar-

ring, and depending on the extent of the Descemet break, it can save patients from penetrating keratoplasty.

### 4.1. What Was Known before

Spontaneous resolution of corneal hydrops is a lengthy process and associated with discomfort and corneal scar.

Approximation pre-Descemet or full thickness suturing with intracameral air or gas is effective in the management of corneal hydrops.

### 4.2. What This Paper Adds

Full-thickness corneal suturing without leaving any gas in the anterior chamber is a simple and safe method to treat acute hydrops, which can be performed in the absence of intraoperative OCT.

**Supplementary Materials:** The following supporting information can be downloaded at: https://www.mdpi.com/article/10.3390/vision7010010/s1, Video S1: Surgical Procedure.

**Author Contributions:** Conceptualization, M.A.N. and Z.A.; methodology, M.A.N. and Z.A; software, Z.A., R.M.; validation, Z.A., R.M., M.A.N.; formal analysis, Z.A.; investigation, Z.A. and R.M.; resources, R.M., M.A.N.; data curation, Z.A. and R.M.; writing—original draft preparation, Z.A. and R.M.; writing—review and editing, M.A.N., Z.A. and R.M.; visualization, M.A.N.; supervision, M.A.N.; project administration, R.M.; funding acquisition: not applicable. All authors have read and agreed to the published version of the manuscript.

**Funding:** No funding was available for this case repot.

**Institutional Review Board Statement:** Ethical review and approval were waived for this study as this study was an audit of cases treated with the similar technique.

**Informed Consent Statement:** Informed consent was obtained from all subjects involved in the study.

**Data Availability Statement:** Not applicable.

**Conflicts of Interest:** The authors declare no conflict of interest.

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
