# Peer review of "Full-Thickness Compressive Corneal Sutures with Removal of Anterior Chamber Air Bubble in the Management of Acute Corneal Hydrops"

_2411-5150, 2022_

Round 1

Reviewer 1 Report

The authors present a study to assess the effect of full thickness corneal suturing as solo treatment in the management of acute hydrops. Five patients with acute hydrops received full thickness corneal sutures. Complete resolution of corneal oedema was observed between 1-2 weeks postoperatively with no complications. The data seem to be thoroughly documented. However, the authors do not present a new information, except of the removal of the air bubble after corneal suturing. New other treatment options such as mini-DMEK to treat corneal hydrops are missing. Therefore, there is a lack of some important literature. Some important issues, such as corneal scaring due to Descemet membrane perforation or prolonged corneal edema after removal of the air bubble are not discussed. Please find my comments/questions attached:

Major revisions:

11.    Introduction: Please mention keratoconus as one reason for corneal hydrops.

22.   Introduction: Nowadays, “conventional management” of corneal hydrops is not conservative anymore. “Formerly management” would be the better term.

33.  Introduction: Mini-DMEK was previously presented by the Professor Bachmann/Cursiefen (Cologne group) with good results. This should be additionally mentioned as a treatment option.

44.   Results: Two out of five patients showed no significant improvement in visual acuity and one patient did not return to the follow-up visit. Therefore, only two out of five patients improved in visual acuity. Please compare these results to other treatment options such as pre-DM “Muraine” sutures or mini-DMEK.

55.    Discussion: Your “traditional” treatment recommendation in the discussion section does not correspond to the introduction.

66.  Discussion: Nowadays, penetrating keratoplasty is not recommended as treatment of corneal hydrops. Please correct: “In the past, some clinicians performed … “

77.   Discussion: I agree, that by removing of the anterior chamber air bubble there is no risk of pupillary block. However, cataract can also induced by the anterior segment surgery itself and an additional air or gas bubble can help to fasten reducing the corneal edema. Therefore, it may be a better option to perform an iridotomy and leave an air bubble in the anterior chamber after the corneal suturing. This should be discussed, otherwise a comparison of the methods should be performed.

88.     Discussion: Please discuss the risk of additional corneal scaring due to Descemet membrane perforation by full thickness corneal suturing. It may be an additional risc of worsening the scars due to the DM perforation.

99.   Discussion: A comparison of the visual acuity is difficult, because of the high keratometry values in keratoconus patients. We do not know, if the visual acuity is reduced due to corneal scars or high keratometry values. The best corrected visual acuity can only be tested by RGPCL. However, some of your patients were not keen on RGPCL. Please discuss this issue.

110.  Discussion: Please discuss and compare mini-DMEK as one treatment option

111.  Conclusion: The presented technique does not save the patient from a keratoplasty. A keratoplasty can be performed after months and resolution of the corneal edema to remove the corneal scar and improve the visual acuity. However, an emergency keratoplasty in no option nowadays and should not be mentioned as “other reported option”. It only can be mentioned as “performed in the past”.

112.  “What this paper adds:” Full thickness corneal suturing were reported by Mohebbi et al., therefore this is not new. The removement of the air bubble need to be added here or the paragraph need to be changed.

113.  The only new information of the paper is the removal of the air bubble. Therefore, it should be considered to change the title of the paper to: “Full-thickness compressive corneal sutures without leaving an air bubble in the anterior chamber as treatment of acute corneal hydrops.” or: “Full-thickness compressive corneal sutures with removal of the anterior chamber air bubble as treatment of acute corneal hydrops.”

Author Response

The authors present a study to assess the effect of full thickness corneal suturing as solo treatment in the management of acute hydrops. Five patients with acute hydrops received full thickness corneal sutures. Complete resolution of corneal oedema was observed between 1-2 weeks postoperatively with no complications. The data seem to be thoroughly documented. However, the authors do not present a new information, except of the removal of the air bubble after corneal suturing. New other treatment options such as mini-DMEK to treat corneal hydrops are missing. Therefore, there is a lack of some important literature. Some important issues, such as corneal scaring due to Descemet membrane perforation or prolonged corneal edema after removal of the air bubble are not discussed. Please find my comments/questions attached:

Major revisions:

  1. Introduction: Please mention keratoconus as one reason for corneal hydrops.

Response: This is now changed in the last sentence of first paragraph on page 4 to:

“Vernal keratoconjunctivitis (VKC), steeper keratometry, atopy, Down syndrome, and eye rubbing are important risk factors for corneal ectasia like keratoconus where corneal hydrops can develop.2,3

  1. Introduction: Nowadays, “conventional management” of corneal hydrops is not conservative anymore. “Formerly management” would be the better term.

Response: This is now changed in second sentence in second paragraph on page 4 to: “Formerly management of hydrops was often conservative as it can resolve spontaneously with extensive scarring.”

  1. Introduction: Mini-DMEK was previously presented by the Professor Bachmann/Cursiefen (Cologne group) with good results. This should be additionally mentioned as a treatment option.

Response: The following sentence is now edited in 3rd paragraph on page 4:

“These include injection of intra-cameral air or gas with or without compressive sutures, pre-Descemet suturing8, anterior segment ocular coherence tomography (ASOCT) guided drainage of intra-stromal fluid.9,10 amniotic membrane transplant,11 mini-Descments’ membrane endothelial keratoplasty (DMEK),12 penetrating keratoplasty and in the rare event of corneal perforation application of tissue adhesives.6

  1. Results: Two out of five patients showed no significant improvement in visual acuity and one patient did not return to the follow-up visit. Therefore, only two out of five patients improved in visual acuity. Please compare these results to other treatment options such as pre-DM “Muraine” sutures or mini-DMEK.

Response: Thank you for your comment. Following full-thickness suturing, the vision improved in four patients and was unknown in one patient, who did not return for further follow-up. In cases 1 and 4, BCVA improved to the pre-hydrops level. In case 2 BCVA improved from HM at presentation to 6/36 post-treatment, although the VA was one line lower than the pre-hydrops level. And in case 3, BCVA improved from HM at presentation to CF post-suturing. Therefore, all patients experienced visual improvement, although it did not reach the pre-hydrops BCVA in two patients. Maurine’s study has compared the BCVA at presentation with post-suturing BCVA, which also showed a significant improvement. Furthermore, two of their patients underwent PK, so the 6-month post-suturing BCVA does reflect not only the suturing outcome but also the added PK outcome. In general, how the vision improves after any of these treatment modalities mainly depends on the location and extent of DM tear and stromal scarring, which varies from patient to patient, and you need a large sample size to be able to compare the visual outcome using different treatment methods. The paragraph is amended as below:

“Regarding the visual outcome, post-procedure BCVA compared to the BCVA at presentation improved in four patients and was unknown in one patient who did not attend the follow-up. Two of our patients achieved their pre-hydrops visual acuity; one patient experienced a one-line drop on the Snellen chart, and another patient, with extensive involvement of the central cornea, had a significant drop on the Snellen chart. Cherif et al. also compared the BCVA at presentation with post-suturing BCVA, which showed a significant improvement. Furthermore, two of their patients required PK once the hydrops settled. Although the visual outcome is not provided in all patients, the graft patients achieved BCVA of 20/20 and 20/32 and another non-grafted eye achieved BCVA of 20/20. Therefore, the 6-month post-suturing BCVA in their study reflects not only the suturing outcome but also the added PK benefit. Likewise, the visual outcome in mini-DMEK technique is not comparable to our study as they provided unaided visual acuity in their three patients, which is understandably poor in eyes with keratoconus regardless of hydrops. Nevertheless, a critical factor determining the final visual outcome is the extent and location of DM break and corneal oedema. The involvement of the central cornea will have a remarkable effect on the final visual outcome. In contrast, paracentral or peripheral DM break and oedema's impact on final vision will be less significant. Some previous studies reported no significant difference in visual outcomes between their intervention and control groups.14,16 Instead of comparing the final visual acuity between two groups, we suggest comparing each patient's pre-hydrops and post-treatment/observation visual acuity to provide a more accurate result.”

  1.   Discussion: Your “traditional” treatment recommendation in the discussion section does not correspond to the introduction.

Response: The following sentences are added to the first paragraph on page 8:

Conventional management of acute hydrops mainly focused on providing symptomatic relief. This included topical sodium chloride eye drops to decrease corneal oedema, cycloplegic eye drops to address the photosensitivity, bandage contact lenses to help with the pain, aqueous suppressants to lower the intraocular pressure and encourage corneal dehydration, and topical steroids and non-steroids to alleviate the inflammation and topical antibiotics to prevent the secondary infection. 1,4,6

  1. Discussion: Nowadays, penetrating keratoplasty is not recommended as treatment of corneal hydrops. Please correct: “In the past, some clinicians performed … “

Response: The sentence is changed in first paragraph on page 8 as below:

Emergency penetrating keratoplasty used to be an alternative surgical option to manage acute hydrops (1). However, this is not the recommended treatment at present, as no report has been published on the survival of penetrating keratoplasty in acute hydrops, and corneal transplant in an inflamed eye is associated with a significantly higher risk of rejection and failure.(13)

  1.  Discussion: I agree, that by removing of the anterior chamber air bubble there is no risk of pupillary block. However, cataract can also induced by the anterior segment surgery itself and an additional air or gas bubble can help to fasten reducing the corneal edema. Therefore, it may be a better option to perform an iridotomy and leave an air bubble in the anterior chamber after the corneal suturing. This should be discussed, otherwise a comparison of the methods should be performed.

Response: thank you for your comment. Combining the full-thickness suturing and anterior chamber gas bubble is a technique which was tried in previous studies. They did not notice any pupillary block in their cohort of 13 patients. However, another study with a larger sample size of 62 eyes, noticed a 16% rate of pupillary block. To prevent pupillary block, either PI should be performed, with potential side effects of hyphaema, IOP spike, dysphotopsia,, etc or post-op cycloplegics should be used as long as there is gas in the AC. The purpose of publishing this case series was to clarify that full-thickness suturing alone, without leaving any AC gas bubble, which requires PI to prevent pupillary block and increases the risk of cataract, addresses acute hydrops. Based on our report, the same outcome can be achieved without the extra risk of pupillary block or cataract.

The paragraph is amended as below:

“Most of the previous reports combined this technique with the injection of air or gas into the anterior chamber. Although some studies with a small number of patients did not come across the pupillary block (3), others with larger sample size noticed up to 16% rate of pupillary block in their patients (16), as it is already understood very well.(18) To prevent this complication, either prophylactic peripheral iridotomy (PI) is to be performed, which is not only difficult to perform via oedematous cornea with poor visibility but also associated with potential complications like hyphaema(19), IOP spike (19), dysphotopsia (20) and the like, or topical cycloplegic drops should be used post-operatively, for as long as the gas is present in the AC. However, with our technique, there is no requirement for prophylactic PI or post-op cycloplegic agents. In all five reported cases, we achieved good anatomical outcomes without leaving any air bubbles in the anterior chamber. “

  1.    Discussion: Please discuss the risk of additional corneal scaring due to Descemet membrane perforation by full thickness corneal suturing. It may be an additional risc of worsening the scars due to the DM perforation.

Response: corneal suturing, either full-thickness (Mohebbi’s study) or pre-Descemet (Maurine’s study) suturing, is associated with scarring. This is secondary to fibrosis which is induced by the stromal keratocytes without any intervention secondary to perforation of the Descemet membrane via full-thickness suturing. However, despite scarring, the vision improved in patients of both studies. To accelerate the resolution of corneal oedema and avoid diffuse scarring, the risk of suture-related linear scarring is acceptable. 

The following sentences are added to 2nd paragraph on page 12 in the discussion:

“Also placing corneal sutures can be associated with suture-tract linear scarring. This was noticed with both pre-Descemet (8) and full-thickness suturing (3). We also came across this complication, however, similar to the previous studies it did not appear to cause significant visual problem and good vision achieved with RGPCL.”

  1.  Discussion: A comparison of the visual acuity is difficult, because of the high keratometry values in keratoconus patients. We do not know, if the visual acuity is reduced due to corneal scars or high keratometry values. The best corrected visual acuity can only be tested by RGPCL. However, some of your patients were not keen on RGPCL. Please discuss this issue.

Response: That is correct, RGPCL trial can define whether the reduced vision is secondary to irregular astigmatism or scarred cornea. Patients 1 and 2 have never been interested in RGPCL as they had good vision in their fellow eye, therefore, their vision was simply compared to the pre-hydrops level. The last sentence of the response to the above comment addresses this question.

  1. Discussion: Please discuss and compare mini-DMEK as one treatment option

Response: the below paragraph is added as 2nd paragraph on page 10 to the discussion:

“Bachmann et al. reported mini-DMEK technique to treat three patients with acute hydrops (12). The procedure was successful in two patients and needed repeating in one patient. To apply this technique iOCT is necessary due to the poor visibility, a donor graft is required on an urgent basis, and the AC needs to be filled with gas. While iOCT is not readily available in all centres and donor tissue shortage is another concern in many countries, the use of anterior chamber gas can be associated with potential complications like pupillary block and cataract.”

  1. Conclusion: The presented technique does not save the patient from a keratoplasty. A keratoplasty can be performed after months and resolution of the corneal edema to remove the corneal scar and improve the visual acuity. However, an emergency keratoplasty in no option nowadays and should not be mentioned as “other reported option”. It only can be mentioned as “performed in the past”.

Response: Not necessarily all patients will need a transplant in the future. If the DM break is not massive and there is no diffuse scarring affecting the visual axis, patients can continue with RGPCL to achieve good vision. This technique saved three of our patients from penetrating keratoplasty. Only one patient with massive Descemet tear needed PK 5 months post-suturing, while the other patients are doing well with glasses or RGPCL. Similarly, in Maurine’s study their technique saved 5 patients from PK, while 2 patients needed PK a few months later. Emergency PK is removed from the options in the last sentence on page 12 as below.

“This technique prevents prolonged discomfort and corneal oedema, leading to vascularisation and scarring, and depending on the extent of the Descemet break, it can save patients from a penetrating keratoplasty.”

  1. What this paper adds:” Full thickness corneal suturing were reported by Mohebbi et al., therefore this is not new. The removement of the air bubble need to be added here or the paragraph need to be changed.

Response: the paragraph is changed as below:

  • Full-thickness corneal suturing without leaving any gas in the anterior chamber is a simple and safe method to treat acute hydrops, which can be performed without intraoperative OCT.

  1. 13.  The only new information of the paper is the removal of the air bubble. Therefore, it should be considered to change the title of the paper to: “Full-thickness compressive corneal sutures without leaving an air bubble in the anterior chamber as treatment of acute corneal hydrops.” or: “Full-thickness compressive corneal sutures with removal of the anterior chamber air bubble as treatment of acute corneal hydrops.”

Response: the title is changed as “Full-thickness compressive corneal sutures with removal of anterior chamber air bubble in the management of acute corneal hydrops”

Reviewer 2 Report

The authors present a case series of five patients assessing the effectiveness of full-thickness corneal suturing as a treatment option in the management of acute hydrops. They present compressive corneal suturing perpendicular to the descemet membrane tear (highlighted by an intracameral air bubble injection) followed by intracameral and topical antibiotics with topical corticosteroids postoperatively as a treatment option with good results.  

Major Strengths: Well-written manuscript. Excellent description of the technique.

Major Weaknesses: The inherent limitations secondary to the nature of a case series. There are a relatively low number of patients included.

Specific issues that need to be addressed by author(s): I would include keratoconus and pseudophakic bullous keratopathy as risk factors for the development of acute hydrops.  I assume the video provides an excellent visual aid – still, possibly a figure/graphic demonstrating the technique would also be beneficial to the reader. I would add to the discussion some mention that this management option does carry with it the inherent risk of anterior chamber entry (infection, damage to adjacent structures) that more conservative approaches such as strictly medical/topical therapy avoid.

Final thoughts: Ultimately, I find it hard to claim full-thickness corneal suturing as the ‘sole’ treatment modality in the manuscript, as an intracameral air bubble was placed and the patient was also treated with medical therapy (intracameral cefuroxime, and topical antibiotic and corticosteroid drops). I understand what the authors are proposing; however, sole therapy in my opinion disregards the medical treatment applied and suggests only a surgical intervention is necessary. Despite the relatively low ‘n’ of patients reported, as the authors suggest, this treatment may be an excellent option in the appropriate patient.

Author Response

The authors present a case series of five patients assessing the effectiveness of full-thickness corneal suturing as a treatment option in the management of acute hydrops. They present compressive corneal suturing perpendicular to the descemet membrane tear (highlighted by an intracameral air bubble injection) followed by intracameral and topical antibiotics with topical corticosteroids postoperatively as a treatment option with good results.  

Major Strengths: Well-written manuscript. Excellent description of the technique.

Response: Thank you for your positive feedback.

Major Weaknesses: The inherent limitations secondary to the nature of a case series. There are a relatively low number of patients included.

*Response: that is true. The small number of patients is mentioned when discussing the visual outcome of this procedure.

Specific issues that need to be addressed by author(s):

I would include keratoconus and pseudophakic bullous keratopathy as risk factors for the development of acute hydrops.  

Response: keratoconus is added to the risk factors as below:

Vernal keratoconjunctivitis (VKC), steeper keratometry, atopy, Down syndrome, and eye rubbing are important risk factors for corneal ectasia like keratoconus where corneal hydrops can develop.(2, 3)

I assume the video provides an excellent visual aid – still, possibly a figure/graphic demonstrating the technique would also be beneficial to the reader.

Response: a figure to demonstrate the stages of hydrops suturing is added as recommended.

 I would add to the discussion some mention that this management option does carry with it the inherent risk of anterior chamber entry (infection, damage to adjacent structures) that more conservative approaches such as strictly medical/topical therapy avoid.

Response: the below paragraph is added to the discussion

Although we did not encounter any complications, when placing full-thickness corneal suture, care should be taken not to damage the adjacent structures. Also, to avoid infection, the use of intracameral antibiotics is highly recommended.  

Final thoughts: Ultimately, I find it hard to claim full-thickness corneal suturing as the ‘sole’ treatment modality in the manuscript, as an intracameral air bubble was placed and the patient was also treated with medical therapy (intracameral cefuroxime, and topical antibiotic and corticosteroid drops). I understand what the authors are proposing; however, sole therapy in my opinion disregards the medical treatment applied and suggests only a surgical intervention is necessary. Despite the relatively low ‘n’ of patients reported, as the authors suggest, this treatment may be an excellent option in the appropriate patient.

Response: the title is changed as below:

“Full-thickness compressive corneal sutures with removal of anterior chamber air bubble in the management of acute corneal hydrops”

Round 2

Reviewer 1 Report

All concerns are adequately answered after the first revision.

Author Response

Many thanks for your comments which helped improve the manuscript.

No further changes are made to the manuscript

Kind regards,

Mayank A. Nanavaty